# Effect of a *Bacillus*-Based Probiotic on Performance and Nutrient Digestibility When Substituting Soybean Meal with Rapeseed Meal in Grower–Finisher Diets

**DOI:** 10.3390/ani13193067

**Published:** 2023-09-29

**Authors:** Marta I. Gracia, Guillermo Cano, Patricia Vázquez, Lea H. B. Hansen

**Affiliations:** 1Imasde Agroalimentaria, S.L. C/Nápoles 3, 28224 Pozuelo de Alarcón, Spain; gcano@e-imasde.com (G.C.); pvazquez@e-imasde.com (P.V.); 2Chr. Hansen A/S, Animal and Plant Health and Nutrition, Boege Allé 10-12, 2970 Hoersholm, Denmark; dkleha@chr-hansen.com

**Keywords:** *B. subtilis*, *B. licheniformis*, pigs, crude protein digestibility, rapeseed meal

## Abstract

**Simple Summary:**

The objective of this study was to test the hypothesis that *B. subtilis* and *B. licheniformis* supplementation in a negative control diet (in comparison to a standard control diet) has the potential to improve performance and nutrient digestibility of growing–finishing pigs. Three fattening pig groups were used: standard diet, negative control diet (5% soybean meal replaced by 5% rapeseed meal), or the negative control diet supplemented with a probiotic. The use of a probiotic preparation containing specific *Bacillus* strains as a feed additive for growing–finishing pigs resulted in improved growth performance and faecal digestibility, and was able to counteract the lower nutritional level of a negative control diet.

**Abstract:**

The objective of the present study was to test the hypothesis of *B. subtilis* and *B. licheniformis* supplementation to a negative control diet in comparison to a standard control diet, had the potential to improve the performance and nutrient digestibility of growing–finishing pigs. For this purpose, 384 fattening pigs of 85 d of age were allotted to three treatments: a standard diet, a negative control (NC) diet (5% soybean meal replaced by 5% rapeseed meal), or a NC diet + probiotic. After reaching a body weight of approximately 110 kg, all animals going to the slaughterhouse (87% of total pigs) were selected to measure carcass quality. Moreover, the apparent total tract digestibility of protein was evaluated at the end of the grower period. The results of this study indicate that supplementation of the tested *Bacillus*-based probiotic significantly improved average daily gain (ADG, +14.6%) and Feed:gain ratio (F:G, −9.9%) during the grower phase compared to the NC diet. The improvement observed during the grower phase was maintained for the whole fattening period (ADG, +3.9%). Probiotic supplementation significantly improved the total apparent faecal digestibility of dry matter and crude protein in pigs at the end of the grower period. The improvements observed with the additive tested could indicate that supplementation of the *Bacillus*-based probiotic was able to counteract the lower level of crude protein and standardised ileal digestible amino acids in the NC diet by means of improved protein digestibility.

## 1. Introduction

*Bacillus*-based probiotics have been reported previously to improve the performance of growing–finishing pigs by means of increasing the growth rate in growing pigs [1,2,3,4], in finishing pigs [5] and in growing–finishing pigs [6], improving the feed:gain (F:G) ratio in growing–finishing pigs [7] and improving the growth and F:G ratio in wean to finish pigs [8,9] and in growing–finishing pigs [10,11,12,13]. The activity of probiotics is influenced by the composition of the diet [14] and variations in dietary protein supply, possibly affecting microbial composition in the gut [15,16]. The performance of broiler chicks and turkey poults was improved via direct-fed microbial inclusion in reduced-fat diets, which was associated with increased energy digestibility [17]. In this sense, in addition to improved performance, several studies suggested that dietary addition of *Bacillus* spp. could lead to increased protein digestibility [1,11,18] and protein and energy digestibility in pigs [9,10,11,19].

Replacing soybean meal (SBM) with protein sources from European agricultural systems, such as as rapeseed meal (RSM), could potentially be a viable cheaper and eco-friendly alternative to imported SBM protein source in pig diets and could contribute to more sustainable pork production [20]. Furthermore, the use of RSM instead of SBM in pig diets could be a viable tool for markedly reducing the negative impact on climate change [21]. Previous studies have shown that SBM can be partially replaced with RSM without affecting the performance of growing–finishing pigs [22,23,24]. However, in those studies, a general adjustment in the composition and a supplementation of amino acids (AA) is normally performed. According to the results of previous studies [25,26], *Bacillus* spp. enhanced the development and activities of digestive enzymes in the gastrointestinal tract, which was associated with an increase in digestibility of some AA in weaning pigs [27].

Therefore, the objective of the present study was to test the hypothesis that *B. subtilis* and *B. licheniformis* supplementation in a negative control diet replacing SBM with RSM in comparison to a standard control diet had the potential to improve the performance and nutrient digestibility of growing–finishing pigs.

## 2. Materials and Methods

### 2.1. Animals, Diets and Housing

A total of 384 fattening pigs [(Landrace × Large White) × Pietrain, 85 d of age, average initial body weight (BW) 29.7 ± 4.7 kg] were randomly allotted to 3 treatments (Control (C) standard diet, Negative Control (NC) diet, and Probiotic (NC + Probiotic)), so that each treatment started with the same initial BW. There were 64 male and 64 female pigs per treatment, allocated in groups of 8 pigs in two adjacent rooms. Experimental diets were fed in 2 phases: grower (85–125 d of age) and finisher (125–187 d of age; Table 1). The C diets were formulated to represent standard commercial grower–finisher diets containing standard levels of energy and protein [28]. The NC diets were formulated by replacing 5% SBM with 5% RSM. The consequence of this replacement was a lower level of crude protein (CP, −4.3%) and standardised ileal digestible (SID) AA (−3%). The third experimental treatment consisted of the NC diet plus 400 mg/kg of a *Bacillus*-based probiotic. The *Bacillus* product comprised a mixture of spray-dried spore-forming *Bacillus licheniformis* (DSM5749) and *Bacillus subtilis* (DSM5750) at a minimum concentration of 3.25 × 10^9^ viable spores/g of product. The experimental diets were fed in a pellet (4.5 mm) form from 85 d of age to slaughter at 187 d of age.

The analysed chemical composition of diets fed during the experiment is presented in Table 2. Pigs were housed in a fattening unit with 100% slatted concrete floor pens (2.6 × 2.6 m). All pens were equipped with a self-feeder and a nipple drinker to allow ad libitum access to feed and water.

### 2.2. Sample Preparation and Measurements

The pigs were weighed individually at the start of the trial and at days 40 and 102 (at the end of the trial). Growth performance in terms of average daily gain (ADG) was calculated individually. Total feed intake per pen was measured at the end of each feeding phase. Average daily feed intake (ADFI) was calculated per pen by dividing the total feed intake for each feeding phase and for the whole trial between the total days of pigs alive + the total days of pigs dead at the end of the period. The Feed:gain ratio was calculated per pen for each feeding phase and for the whole trial by dividing the mean ADG of the pen between the ADFI. Dressing yield was determined individually on the basis of final BW and hot carcass weight obtained from the Meat Processing Plant. After reaching a body weight of approximately 110 kg, all animals going to the slaughterhouse (87% of total pigs) were selected to measure carcass quality. The duration of fasting time before slaughter was 24 h. On the slaughter line, the lean meat percentage in the carcass, the loin thickness, backfat in the carcass and subcutaneous ham fat were non-invasively ultrasonically measured (AutoFom, SFK Technology, Herlev, Denmark). Before cooling, the hot carcass weight was determined with an accuracy of 100 g. To evaluate the effect of dietary treatments on the apparent total tract digestibility (ATTD), acid-insoluble ash (diatomaceous earth, Celatom Clarcel DIC/M, Manuel Riesgo, Madrid, Spain) was added to the grower diets at 5 g/kg as an inert, indigestible indicator. The pigs were fed the diets containing acid insoluble ash during the entire grower period, and fresh faecal grab samples were collected randomly from three pigs from each pen during the last three days of the grower period. The faecal samples were pooled within pen, dried in a forced air-drying oven at 60 °C for 72 h, and ground in Wiley mill (Thomas Model 4 Wiley Mill, Thomas Scientific, Swedesboro, NJ, USA) using a 1 mm screen and used for chemical analysis.

The ATTD of CP and dry matter (DM) in the assay diets was calculated according to the following equation:ATTD = [1 − (N_F_/N_D_) × (AIA_D_/AIA_F_)] × 100
where ATTD is the ATTD of CP or DM in the assay diet (%), N_F_ is the nutrient content in faeces (g/100 g DM), N_D_ is the nutrient content in the assay diet (g/100 g DM), AIA_D_ is the inert marker content in the assay diet (g/100 g DM), and AIA_F_ is the inert marker content in faeces (g/100 g DM).

### 2.3. Laboratory Analysis

Feed samples of each experimental diet were collected from every batch, then pooled, and sub-samples were analysed for proximate chemical compositions (AOAC methods). Experimental diets were analysed in triplicate for ash (method 942.05), ether extract (method 920.39), crude fibre (method 962.09), calcium (method 968.08) and phosphorus (method 965.17) according to [29]. Dry matter (method 930.15), CP (method 990.03) and acid-insoluble ash [30] of diets and faecal matter were also determined in triplicate. *Bacillus* spores in feed were enumerated according to Chr. Hansen SOP-03532: “Bacillus Recovery Program: Enumeration of *Bacillus* strains used in animal feed, water and seed samples”, based on and in compliance with the European standard EN-15784:2020 (TC327 WI00327117:2021) “Animal feeding stuffs—Isolation and enumeration of *Bacillus* strains used as feed additive.”.

### 2.4. Statistical Analysis

Data generated in this experiment were analysed via ANOVA, using the GLM procedure of IBM SPSS Statistics for Windows Version 29.0. (IBM Corp., Armonk, NY, USA) for a randomized complete block design evaluating the experimental treatment. Initial BW was used as a covariate for zootechnical performance and sex and room were included as fixed effects. The individual data were used as the experimental unit for BW, ADG and slaughterhouse measurements, while the pen was used as the experimental unit for ADFI, F:G ratio and ATTD. Tables show least square means. Single contrasts were used for pair-wise comparison of experimental treatments. Also, the 95% confidence intervals of the difference between the negative control group and the probiotic supplemented group were calculated. Probability values of *p* ≤ 0.05 were considered significant, whereas 0.05 < *p* ≤ 0.10 was considered as a tendency.

## 3. Results

### 3.1. Growth Performance and Carcass Evaluation

Replacing SBM with RSM (C vs. NC) had a negative impact on performance, reducing the BW of pigs after 40 days on trial (53.8 vs. 51.8 kg, *p* = 0.0340) and numerically at the end of the trial (113.9 vs. 111.5 kg, *p* = 0.1273) and reducing growth (ADG) during the grower phase (601 vs. 552 g/d, *p* = 0.0340) (Table 3). No significant differences were observed between C and NC pigs in terms of feed intake or F:G ratio.

Probiotic supplementation of the NC diets increased BW after 40 days of trial (*p* = 0.0004) and at the end of the trial (*p* = 0.0390) (Table 2). In addition, with 95% confidence, the BW after 40 days of trial of probiotic-supplemented pigs was between 1.5 and 5.2 kg heavier than BW of NC pigs. Also, with 95% confidence, the BW at the end of the fattening period of probiotic-supplemented pigs was between 0.2 and 6.3 kg heavier than the BW of NC pigs.

Probiotic supplementation to the NC diets increased the growth of pigs (ADG) during the grower period (*p* = 0.0004). With 95% confidence, the ADG of probiotic-supplemented pigs increased between 38.15 and 130.4 g/d over the NC pigs. Probiotic supplementation to the NC diets decreased the F:G ratio of pigs during the grower period (*p* = 0.0254). With 95% confidence, the F:G ratio of probiotic-supplemented pigs improved between 0.03 and 0.45 units over the NC pigs.

Zootechnical improvement with probiotic supplementation was not detected during the finisher phase. However, when evaluating the global performance period, the probiotic supplementation to the NC diets increased the ADG (*p* = 0.0390) and decreased the F:G ratio of pigs (*p* = 0.0280). In addition, with a 95% confidence, the ADG for the whole fattening period of probiotic-supplemented pigs was between 1.6 and 61.3 g/d higher than ADG of NC pigs and F:G ratio of probiotic-supplemented pigs was improved between 0.008 and 0.133 units over the NC pigs.

No significant differences in performance were observed when comparing the NC + Probiotic treatment and C treatment, indicating that probiotic-supplemented pigs performed to the same level as C pigs.

No significant differences between treatments were observed in global mortality, with a total of 2, 2 and 4 pigs dying while receiving C, NC and Probiotic treatments, respectively.

No significant differences between treatments were observed for the slaughterhouse measurements at the end of the trial: carcass weight, dressing percentage, back-fat thickness and lean meat percentage (Table 4).

### 3.2. Apparent Nutrient Digestibility

Probiotic supplementation of NC diets significantly improved the total tract digestibility of DM and CP, reaching the values showed by the C pigs (Table 5).

## 4. Discussion

When SBM is partially or totally replaced with RSM in the diet, a general adjustment in the composition and a supplementation (e.g., AA) may be necessary in order to maintain the performance. However, in our study, the diets were formulated only to keep the ratio between AA, but not to maintain the CP. Consequently, when SBM was partially substituted with RSM, the NC feed went below what is recommended (∼4.3% less in calculated CP) and the reduced level of AA was 3% lower in calculated SID AA. As expected, the performance of NC pigs was impaired compared to that of C animals, probably as a consequence of a generally lower protein digestibility, especially precaecal [31,32].

The tested probiotic improved growth and feed efficiency over the NC in the grower phase of growth. Rybarczyk et al. [13], reported lower F:G, higher ADG and shorter fattening period in pigs supplemented with the same *Bacillus*-based probiotic compared to pigs from the control group. Balasubramanian et al. [11] reported that supplementation with a commercially available *Bacillus*-based probiotic, containing *B. coagulans*, *B. lichenformis* and *B. subtilis*, effectively improved growth performance and F:G without affecting feed intake. Additionally, Chen et al. [2] found that dietary supplementation with a *Bacillus*-based probiotic (*B. subtilis* and *B. coagulans*) effectively improved the growth performance in growing pigs. Alexopoulos et al. [8] also observed a significant improvement in growth performance when feeding finishing pigs a diet supplemented with the same *Bacillus*-based probiotic as investigated in this study. Moreover, Cui et al. [33] concluded that the addition of *B. subtilis* improved ADG and ADFI and decreased F:G. In most of these studies, a positive effect of *Bacillus*-based probiotics on the performance of growing–finishing pigs was observed. Meng et al. [10] reported increased ADG during both the growing and finishing phases; however, F:G was not significantly improved in the finishing phase. Similar effects were observed in the present study, which may confirm the idea that older pigs are better able to resist intestinal disorders and thus obtain less benefit from probiotics.

The improved BW and growth observed in our trial due to dietary supplementation of *Bacillus*-based probiotic during the grower period might be associated, at least in part, with the improvement in CP and DM digestibility shown by the probiotic during the same period. These results could indicate that supplementation of the *Bacillus*-based probiotic counteracted the lower level of CP% and SID AA in the NC diet by means of improved CP digestibility. Jørgensen et al. [9] reported improvement in ATTD of CP following supplementation with the same *Bacillus*-based probiotic in grower pigs fed an energy-reduced diet, and Balasubramanian et al. [11] revealed that another *Bacillus* probiotic had a significant effect on the digestibility of DM. However, Kaewtapee et al. [34] did not find any improvement in apparent and standardized ileal digestibility of CP and AA in growing pigs fed diets supplemented with *B. subtilis* and *B. licheniformis*. Blavi et al. [19] showed a greater apparent ileal digestibility of total AA in the diet supplemented with *B. amyloliquefaciens* but no effect on digestibility with *B. subtilis* compared to the control diet. On the other hand, Lee et al. [35] reported improvement in ATTD of DM, CP and gross energy when pigs were fed diets supplemented with *B. subtilis* fermentation biomass. Meng et al. [10] suggested that energy and nutrient density influence the effects of probiotics on the gastrointestinal tract, the utilization of nutrients and subsequent pig performance. In our study, the NC diets were formulated by replacing 5% SBM with 5% RSM. A comparison of the protein content of SBM and RSM shows that the latter only contains 75% protein. In general, higher amounts of neutral detergent fibre and acid detergent fibre can be expected in RSM, which is also due to higher amounts of lignin-containing shells [32,36,37]. Thus, the amount and composition of crude fibre are known to have a direct effect on protein digestibility, especially in younger pigs [38,39]. Therefore, it must be expected that replacing SBM with RSM comes with generally lower protein digestibility. The increase in the digestibility of protein observed with the tested *Bacillus*-based probiotic over the NC diet may be a result of the fact that spore-forming Bacilli can synthetize extracellular enzymes, including α-amylase, cellulose, proteases and metalloproteases [40,41,42,43]. *B. subtilis* also secretes α-amylase [44] and it is also possible that *B. subtilis* synthesizes fibre hydrolysing enzymes, such as pectinase and xylanase [25], which may have contributed to increased fermentation of dietary fibre. In addition, the absorptive cells of the villi of broilers supplemented with *B. subtilis* revealed structural changes, including hyperplasia and increased goblet cell population and microvilli height [45].

In the present study, there were no differences between the probiotic-supplemented group and the negative control in terms of carcass weight, dressing percentage, lean meat and backfat thickness. Similar results in terms of carcass weight, dressing yield and lean meat were obtained following supplementation of the same probiotic product by Rybarczyk et al. [13]. However, Alexopoulos et al. [8] reported an improvement in the carcass quality of fatteners supplemented with *Bacillus* in feed. In addition, supplementation of *Bacillus* spp. probiotic improved carcass weight in several studies [11,46,47,48]. Balasubramanian et al. [11], on the other hand, reported no effect on backfat thickness with the supplementation of *Bacillus* spp. probiotic. However, Cui et al. [33] observed increased backfat and *longissimus* muscle area when pigs were supplemented with a probiotic containing *B. subtilis*. These inconsistent results found in carcass quality may be influenced by pig genotype with differences in lean growth and fat deposition, and by the age and BW of pigs at slaughter.

Overall, the probiotic further improved the utilization of protein in the NC diet, having a positive effect on production, lowering F:G ratio and increasing growth. Taking into consideration the reduction of N in the NC diet, as well as the positive effects found when the probiotic was supplemented in the NC diet, replacing SBM with RSM could potentially be a viable cheaper and eco-friendly alternative to imported SBM protein sources in pig diets. Furthermore, the use of RSM instead of SBM in pig diets could be a viable tool for markedly reducing these animals’ negative impact on climate change [21].

## 5. Conclusions

The results of this study indicate that supplementation with the tested *Bacillus*-based probiotic significantly improved growth (ADG, +14.6%) and F:G ratio (F:G, −9.9%) during the grower phase compared to the NC diet. The improvement observed during the grower phase was maintained for the whole fattening period (ADG, +3.9%). Probiotic supplementation significantly improved the ATTD of DM and CP in pigs at the end of the grower fattening period. The improvements observed with the additive tested could indicate that supplementation of the *Bacillus*-based probiotic was able to counteract the lower level of CP% and SID AA in the NC diet by means of improved CP digestibility.

## Figures and Tables

**Table 1 animals-13-03067-t001:** Ingredients and calculated composition of the experimental diets.

Ingredients, g/100 g	Grower, 29–55 kg		Finisher, 55–110 kg	
	Control (C)	Negative Control (NC)	Control (C)	Negative Control (NC)
Barley	46.995	46.595	50.991	50.622
Corn	15.000	15.000	15.000	15.000
Wheat	12.000	12.000	12.000	12.000
Rapeseed meal	--	5.000	--	5.000
Soybean meal	19.578	14.578	16.038	11.038
Animal fat	2.956	3.311	2.651	3.006
Calcium carbonate ^1^	0.943	0.902	0.876	0.835
Monocalcium phosphate	0.912	0.898	0.885	0.871
Salt	0.446	0.443	0.420	0.416
Methionine-OH	0.132	0.116	0.099	0.086
L-lysine (50)	0.538	0.602	0.518	0.587
L-threonine (98)	0.131	0.140	0.117	0.128
L-tryptophan	0.009	0.015	0.005	0.012
Acid Insoluble Ash ^2^	0.500	0.500	--	--
Vitamin and mineral Premix ^3^	0.400	0.400	0.400	0.400
Calculated analysis, g/100 g				
Net energy, kcal/kg	2440	2440	2440	2440
Crude protein	16.40	15.71 (−4.21%)	15.10	14.41 (−4.55%)
Ether extract	4.75	5.14	4.45	4.85
Crude fibre	3.71	4.09	3.74	4.12
Calcium	0.72	0.72	0.68	0.68
Total phosphorus	0.56	0.58	0.54	0.56
Digestible phosphorus	0.29	0.29	0.28	0.28
Total Lys	1.06	1.04	0.97	0.95
SID Lys	0.96	0.9312 (−3.0%)	0.87	0.8439 (−3.0%)
SID Met	0.33	0.32	0.29	0.27
SID Met + Cys	0.58	0.5626 (−3.0%)	0.52	0.5044 (−3.0%)
SID Thr	0.62	0.6014 (−3.0%)	0.56	0.5432 (−3.0%)
SID Trp	0.18	0.1746 (−3.0%)	0.16	0.1552 (−3.0%)

^1^ A total of 400 g/t of calcium carbonate was replaced by the probiotic product in the Probiotic supplemented group. ^2^ Acid-insoluble ash: diatomaceous earth. ^3^ Provided per kilogram of diet: Vitamin A (3a672a): 6500 IU; Vitamin D3 (3a671): 1500 IU; Vitamin E (3a700): 20.0 mg; Vitamin B2: 3.0 mg; Calcium pantothenate (3a841): 10.0 mg; Vitamin B6 (3a831): 1.0 mg; Vitamin B12: 15.0 µg; Nicotinic acid (3a314): 15.0 mg; Betain (3a925): 23 mg; Choline chloride: 50.0 mg; Fe (FeSO_4_·H_2_O 3b101): 100.0 mg; Cu (CuSO_4_·5H_2_O 3b405): 10.0 mg; Mn (MnO 3b502): 60.0 mg; Zn (ZnO 3b603): 100.0 mg; I (KI 3b201): 1.0 mg; Se (Na_2_SeO_3_ 3b801): 0.2 mg. The C diets were formulated to represent standard commercial grower–finisher diets containing standard levels of energy and protein. The NC diets were formulated by replacing 5% soybean meal with 5% rapeseed meal. SID = standardised ileal digestible.

**Table 2 animals-13-03067-t002:** Analysed composition of the experimental diets.

Analysed Composition, g/100 g		Grower, 29–55 kg			Finisher, 55–110 kg		
		Control (C)	Negative Control (NC)	NC + Probiotic	Control (C)	Negative Control (NC)	NC + Probiotic
Moisture		9.04	8.63	8.27	9.39	9.10	8.92
Ash		4.57	4.40	4.47	4.58	4.46	4.40
Crude protein		16.34	15.86	15.75	15.21	14.15	14.25
Crude fat		4.37	5.07	5.24	4.33	4.92	4.99
Crude fibre		4.14	4.49	4.47	4.08	4.24	4.33
Starch		42.36	42.23	42.65	43.30	44.30	44.01
Probiotic, CFU/g	Expected	<1.00 × 10^5^	<1.00 × 10^5^	1.30 × 10^6^	<1.00 × 10^5^	<1.00 × 10^5^	1.30 × 10^6^
	Analysed	<1.00 × 10^5^	<1.00 × 10^5^	1.16 × 10^6^	<1.00 × 10^5^	<1.00 × 10^5^	1.09 × 10^6^

The C diets were formulated to represent standard commercial grower–finisher diets containing standard levels of energy and protein. The NC diets were formulated by replacing 5% soybean meal with 5% rapeseed meal. The third experimental treatment consisted of the NC diet plus 400 mg/kg of a *Bacillus*-based probiotic. CFU = colony-forming units.

**Table 3 animals-13-03067-t003:** Effect of replacing soybean meal with rapeseed meal and of probiotic supplementation on zootechnical performance of pigs.

Parameter	Control (C)	Negative Control (NC)	NC + Probiotic	SEM	*p*-Value	Single Contrasts			95% CI of the Difference NC-Probiotic	
						C vs. Probiotic	NC vs. Probiotic	C vs. NC		
Body weight, kg										
Trial start	29.7	29.7	29.8	0.38	0.9723	0.8225	0.8585	0.9633	−1.16	0.97
After 40 days on trial	53.8	51.8	55.2	0.66	0.0016	0.1412	0.0004	0.0340	−5.21	−1.52
Trial end	113.9	111.5	114.7	1.08	0.1007	0.5831	0.0390	0.1273	−6.26	−0.16
Average daily gain (ADG), g										
Grower phase	601	552	636	16.4	0.0016	0.1412	0.0004	0.0340	−130.37	−38.11
Finisher phase	969	963	961	12.4	0.8922	0.6475	0.9106	0.7290	−32.83	36.81
Global trial	825	801	833	10.6	0.1007	0.5831	0.0390	0.1273	−61.35	−1.59
Average daily feed intake (ADFI), g										
Grower phase	1332	1274	1345	33.2	0.2883	0.7808	0.1398	0.2274	−165.18	24.06
Finisher phase	2380	2336	2333	37.6	0.6203	0.3867	0.9608	0.4138	−104.81	110.08
Global trial	1966	1916	1940	30.6	0.5182	0.5400	0.5942	0.2547	−110.68	64.17
Feed:gain (F:G) ratio										
Grower phase	2.30	2.41	2.17	0.073	0.0798	0.2365	0.0254	0.2706	0.031	0.450
Finisher phase	2.46	2.44	2.43	0.032	0.7143	0.4436	0.8881	0.5309	−0.084	0.096
Global trial	2.40	2.41	2.34	0.022	0.0704	0.0899	0.0280	0.5920	0.008	0.133

The C diets were formulated to represent standard commercial grower–finisher diets containing standard levels of energy and protein. The NC diets were formulated replacing 5% soybean meal by 5% rapeseed meal. The third experimental treatment consisted of the NC diet plus 400 mg/kg of a *Bacillus*-based probiotic. CI = confidence interval; ADG = average daily gain measured individually; ADFI = average daily feed intake; F:G ratio = feed conversion rate. SEM = standard error of the mean. Table shows LSMeans.

**Table 4 animals-13-03067-t004:** Effect of replacing soybean meal with rapeseed meal and of probiotic supplementation on slaughterhouse measurements.

Parameter	Control (C)	Negative Control (NC)	NC + Probiotic	SEM	*p*-Value	Single Contrasts			95% CI of the Difference NC-Probiotic	
						C vs. Probiotic	NC vs. Probiotic	C vs. NC		
n	114	110	113							
BW at end trial, kg	117.1	115.0	117.4	1.10	0.2618	0.8509	0.1327	0.1859	−5.44	0.72
Carcass weight, kg	92.7	90.6	92.3	0.93	0.2301	0.7570	0.1905	0.1055	−4.31	0.86
Dressing percentage, %	79.1	78.9	79.0	0.25	0.8602	0.7696	0.8023	0.5840	−0.81	0.63
Lean meat *, %	62.0	62.3	62.3	0.19	0.4411	0.2480	0.9184	0.2957	−0.56	0.50
Loin thickness *, mm	63.0	63.5	62.2	0.55	0.2477	0.3187	0.0975	0.5019	−0.24	2.85
Back fat carcass *, mm	15.3	15.2	15.1	0.24	0.8421	0.5771	0.6647	0.9059	−0.52	0.82
Min. ham fat *, mm	10.0	9.7	9.8	0.19	0.5470	0.4953	0.6837	0.2782	−0.63	0.42
Subcutaneous ham fat *, mm	20.4	19.6	20.1	0.30	0.1719	0.5769	0.2008	0.0670	−1.38	0.29

The C diets were formulated to represent standard commercial grower–finisher diets containing standard levels of energy and protein. The NC diets were formulated by replacing 5% soybean meal with 5% rapeseed meal. The third experimental treatment consisted of the NC diet plus 400 mg/kg of a *Bacillus*-based probiotic. CI = confidence interval; n = number of observations; BW = body weight. * Parameters estimated by AutoFom.

**Table 5 animals-13-03067-t005:** Effect of replacing soybean meal with rapeseed meal and of probiotic supplementation on the apparent total tract digestibility (ATTD, %) at the end of the grower period.

Parameter	Control (C)	Negative Control (NC)	NC + Probiotic	SEM (n = 16)	*p*-Value
ATTD of DM	82.5 ^ab^	82.0 ^b^	83.3 ^a^	0.25	0.0018
ATTD of CP	81.5 ^a^	79.9 ^b^	81.4 ^a^	0.41	0.0137

The C diets were formulated to represent standard commercial grower–finisher diets containing standard levels of energy and protein. The NC diets were formulated by replacing 5% soybean meal with 5% rapeseed meal. The third experimental treatment consisted of the NC diet plus 400 mg/kg of a *Bacillus*-based probiotic. DM = dry matter; CP = crude protein; SEM = standard error of the mean (n: number of observations). LSMeans with different superscripts within the same row are significantly different (a,b *p* ≤ 0.05); Tukey’s test.

## Data Availability

The data presented in this study are available on request from the corresponding author. The data are not publicly available due to privacy concerns.

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
