# Peer review of "Effect of a *Bacillus*-Based Probiotic on Performance and Nutrient Digestibility When Substituting Soybean Meal with Rapeseed Meal in Grower–Finisher Diets"

_animals, 2023, doi:10.3390/ani13193067_

Round 1

Reviewer 1 Report

The paper entitled „Effect of a Bacillus-based probiotic on performance and nutrient digestibility…” is clear and well-written. However, there are some things that need to be improved:

- Simple summary: it is too brief. Please, extend it a bit.

- Introduction: it is also too brief. Some more information about previous findings would be needed.

- L31: there are too many citations. Please, include the most relevant ones up to 5 or 6.

- L55: define the abbreviations G and F because it is the first time that they are used.

- L57: it is the first time that the abbreviation appears so please, add “crude protein” to CP.

- L58: same as previous comment (this time for SID).

- L92: remove “of protein” after ATTD because also ATTD of DM is shown.

- L100-L106: It is written ATTD of CP but in the formula is written N. In results also ATTD of CP is shown. Could you please clarify if N means nitrogen, that the coefficient 6.25 was used to convert it into CP? Or change in the formula N for CP. On the other hand, iF N in the formula means "nutrient", please clarify it in lines 103-106.

- L110: please check where was the first time that dry matter appeared and the abbreviation was used.

- L111: use only the abbreviation for “crude protein”

- L113-L114: please, remove the methods for DM and CP because that they were already written previously.

- L132: BW at trial end was maybe numerically different but not statistical (p=0.1273). Please, clarify this.

- L171-173: This sentence is not for “Results” but for “Discussion”

- Table 1: correct the superscript in “Calcium carbonate”. On the other hand, the table is a bit confused because it has 4 columns of compositions (4 feed) but also 6. Would it be possible to separate it in 2 tables? Finally, in “analyzed composition”, is the value of CP in the grower diet of the group NC+probiotic correct? From the ingredient composition I don´t understand such high and different value…

- Table 2: Is correct the p-value and the single contrasts in the case of “BW” after 40 days and trial end and of “average daily gain” in grower phase and global trial? They are the same (BW 40 days = ADG grower phase; BW end = ADG global trial). I understand that it would be the same when comparing “BW gain (total)” and “average daily gain”, but not with BW.

Author Response

Dear Reviewer,

Many thanks for the revision of the manuscript. Please, fin below our answers to the received comments.

Kind regards.

Response to Reviewer(s)' Comments to Author:

Reviewer #1.

The paper entitled „Effect of a Bacillus-based probiotic on performance and nutrient digestibility…” is clear and well-written. However, there are some things that need to be improved:

Many thanks for the positive comment and all your help for improving the manuscript. We have taken into account all your suggestions

- Simple summary: it is too brief. Please, extend it a bit. Done.

- Introduction: it is also too brief. Some more information about previous findings would be needed. Done.

- L31: there are too many citations. Please, include the most relevant ones up to 5 or 6. Done.

- L55: define the abbreviations G and F because it is the first time that they are used. Done.

- L57: it is the first time that the abbreviation appears so please, add “crude protein” to CP. Done.

- L58: same as previous comment (this time for SID). Done.

- L92: remove “of protein” after ATTD because also ATTD of DM is shown. Done.

- L100-L106: It is written ATTD of CP but in the formula is written N. In results also ATTD of CP is shown. Could you please clarify if N means nitrogen, that the coefficient 6.25 was used to convert it into CP? Or change in the formula N for CP. On the other hand, iF N in the formula means "nutrient", please clarify it in lines 103-106. Many thanks for the observation. We have reviewed the paragraph and changed to “nutrient” in order to include both DM and CP.

- L110: please check where was the first time that dry matter appeared and the abbreviation was used. Done.

- L111: use only the abbreviation for “crude protein” Done.

- L113-L114: please, remove the methods for DM and CP because that they were already written previously. Done.

- L132: BW at trial end was maybe numerically different but not statistical (p=0.1273). Please, clarify this. Many thanks for the observation. The word “numerically” has been included.

- L171-173: This sentence is not for “Results” but for “Discussion” Many thanks for the observation. The sentence has been moved to the discussion.

- Table 1: correct the superscript in “Calcium carbonate”. Done. On the other hand, the table is a bit confused because it has 4 columns of compositions (4 feed) but also 6. Would it be possible to separate it in 2 tables? The table has been divided. Finally, in “analyzed composition”, is the value of CP in the grower diet of the group NC+probiotic correct? From the ingredient composition I don´t understand such high and different value… Many thanks for the observation. The value of CP has been corrected

- Table 2: Is correct the p-value and the single contrasts in the case of “BW” after 40 days and trial end and of “average daily gain” in grower phase and global trial? They are the same (BW 40 days = ADG grower phase; BW end = ADG global trial). I understand that it would be the same when comparing “BW gain (total)” and “average daily gain”, but not with BW. Many thanks for the observation. We have reviewed all values and they are correct. The P-values for the treatment effect are the same for BW and ADG because initial BW was used as a covariate in the statistical model.

Reviewer 2 Report

The paper submitted by Gracia et al. assessed the effect of Bacillus-based probiotic on performance and nutri- 2 ent digestibility when substituting soybean meal with a rape-seed meal in grower-finisher diets. The authors found that supplementation of the Bacillus-based probiotic could counteract the NC diet's lower CP% and SID AAs level through improved CP digestibility. The manuscript is well organized and written, and I recommend its publication in „Animals Journal. “However, I suggest to consider the following minor comments:

1-     Please expand the full name of all abbreviations mentioned for the first time: SID (line 26, SID line 58).

2-     Line 35: Please add the following sentence: The performance of broiler chicks and turkey poults was improved by direct-fed microbial inclusion in reduced fat diets which was associated with increased energy digestibility. https://doi.org/10.51585/gjvr.2021.4.0024

More detailed comments can be found in the attachment.

 Minor editing of English language required

Author Response

Dear reviewer,

Many thanks for the revision of the manuscript. Please find below our answers to the received comments.

Kind regards

Response to Reviewer(s)' Comments to Author:

 Reviewer #2.

The paper submitted by Gracia et al. assessed the effect of Bacillus-based probiotic on performance and nutrient digestibility when substituting soybean meal with a rape-seed meal in grower-finisher diets. The authors found that supplementation of the Bacillus-based probiotic could counteract the NC diet's lower CP% and SID AAs level through improved CP digestibility. The manuscript is well organized and written, and I recommend its publication in „Animals Journal. “However, I suggest to consider the following minor comments:

Many thanks for the recommendation for publication and for all your help for improving the manuscript. We have taken into account all your suggestions.

1-     Please expand the full name of all abbreviations mentioned for the first time: SID (line 26, SID line 58). Done.

2-     Line 35: Please add the following sentence: The performance of broiler chicks and turkey poults was improved by direct-fed microbial inclusion in reduced fat diets which was associated with increased energy digestibility. https://doi.org/10.51585/gjvr.2021.4.0024. The sentence and reference have been added.

More detailed comments can be found in the attachment.

Simple summary: Too summarized, needs to be expanded. Done.

Abstract: well written however, there are several abbreviations that need to be expanded when mentioned for the first time.

Line 20: Please expand the full name of ADG (average daily gain), Done.

Line 20: Please expand the full name of F:G (Feed: gain ratio). Done.

Line 26: Please expand the full name of CP. Done.

Introduction: Too short, I suggest expanding to the covert he state of the art and the need for this study. The introduction has been expanded.

Please expand the full names of all abbreviations mentioned for the first time: SID (line 26, SID line 58, Line 55 G/D). Done.

Too many citations in some paragraphs (line 31); please reduce them. Done.

Line 35: Please add the following sentence: The performance of broiler chicks and turkey poults was improved by direct-fed microbial inclusion in reduced fat diets, which was associated with increased energy digestibility. https://doi.org/10.51585/gjvr.2021.4.0024. The sentence and reference have been added.

Materials and methods:

L92: please delete –“of protein” after ATTD. Done.

Line 110: Please delete dry matter (already mentioned). Done.

Line 113-114: unnecessary repetition. Done.

Table 1. Analysis composition should be presented in two tables. The table has been divided.

Please expand the full name of CFU (colony forming unit). How do the authors analyze the CFU? It should be mentioned in the materials and methods briefly. Done. The analysis procedure is now included in the materials and methods section.

Results

Lines 171-173: should be used in the discussion part. Many thanks for the observation. The sentence has been moved to the discussion.

Tables should be stand-alone; please explain the treatment/all abbreviations in the footnotes. Done.

Discussion

Line 219: Line 35: Inclusion of direct-fed microbial inclusion in reduced fat diets improved the performance, associated with increased energy digestibility. https://doi.org/10.51585/gjvr.2021.4.0024. We are sorry but we do not agree with this addition as the sentence and reference refer not only to poultry but also to different probiotics (not Bacillus), direct fed microorganism cocktail containing Lactobacillus acidophilus, Lactobacillus casei subsp. rhamnosus, Bifidobacterium bifidum and Enterococcus faecium.

Line 233: In addition, the absorptive cells of the villi revealed structural changes, including hyperplasia and increased goblet cell population and microvilli height (. https://doi.org/10.51585/gjvr.2023.3.0058). The sentence and reference have been added.

Conclusion

Please use the abbreviations of names that have been previously mentioned (i.e., CP). Done.

Round 2

Reviewer 1 Report

The authors did a good job in improving the manuscript according to the reccomendations and I think that it can be published in its present form after correcting two minor things:

- Table 1: the superscript "1" is still not corrected.

- L299: please, change TAFD for ATTD.

Author Response

Dear Reviewer, many thanks for the review. We hve adapted the two minor things pointed out:

Table 1: the superscript "1" is still not corrected.

Sorry, we totally forgot to delete the 1. Now it is deleted.

L299: please, change TAFD for ATTD.

Done.

Kindest regards.